# DKK3, Downregulated in Invasive Epithelial Ovarian Cancer, Is Associated with Chemoresistance and Enhanced Paclitaxel Susceptibility via Inhibition of the β-Catenin-P-Glycoprotein Signaling Pathway

**DOI:** 10.3390/cancers14040924

**Published:** 2022-02-12

**Authors:** Que Thanh Thanh Nguyen, Hwang Shin Park, Tae Jin Lee, Kyung-Mi Choi, Joong Yull Park, Daehan Kim, Jae Hyung Kim, Junsoo Park, Eun-Ju Lee

**Affiliations:** 1Department of Obstetrics and Gynecology, School of Medicine, Chung-Ang University, Seoul 06974, Korea; thanhque2610@gmail.com (Q.T.T.N.); rudal27@hanmail.net (K.-M.C.); 2Department of Obstetrics and Gynecology, Chung-Ang University Health Care System, Hyundae Hospital, Namyangju 12013, Korea; hwangshini@hanmail.net; 3Department of Pathology, School of Medicine, Chung-Ang University, Seoul 06974, Korea; taejlee@cau.ac.kr; 4Department of Mechanical Engineering, Chung-Ang University, Seoul 06974, Korea; jrpark@cau.ac.kr (J.Y.P.); toto594@naver.com (D.K.); 5Department of Radiology, Sanggye Paik Hospital, Inje University College of Medicine, Seoul 01757, Korea; gimage@naver.com; 6Division of Biological Science and Technology, Yonsei University, Wonju 26493, Korea; junsoo@yonsei.ac.kr

**Keywords:** ovarian cancer, DKK3, paclitaxel resistance, β-catenin, P-glycoprotein

## Abstract

**Simple Summary:**

Dickkopf-3 (DKK3) is considered a tumor suppressor as it possesses anti-tumoral properties and is frequently downregulated in various cancers. However, the role of DKK3 in ovarian cancer is not known. In this study, we showed that DKK3 loss occurred in 56.1% of patients with ovarian cancer and that it was significantly associated with poor survival and chemoresistance. Secreted DKK3 possessed anti-tumoral properties and enhanced paclitaxel susceptibility by inhibiting the β-catenin-P-glycoprotein signaling pathway in ovarian cancer. This study revealed promising therapeutic effects of secreted DKK3, which targets paclitaxel-resistant ovarian cancer.

**Abstract:**

Dickkopf-3 (DKK3), a tumor suppressor, is frequently downregulated in various cancers. However, the role of DKK3 in ovarian cancer has not been evaluated. This study aimed to assess aberrant DKK3 expression and its role in epithelial ovarian carcinoma. DKK3 expression was assessed using immunohistochemistry with tissue blocks from 82 patients with invasive carcinoma, and 15 normal, 19 benign, and 10 borderline tumors as controls. Survival data were analyzed using Kaplan–Meier and Cox regression analysis. Paclitaxel-resistant cells were established using TOV-21G and OV-90 cell lines. Protein expression was assessed using Western blotting and immunofluorescence analysis. Cell viability was assessed using the MT assay and 3D-spheroid assay. Cell migration was determined using a migration assay. DKK3 was significantly downregulated in invasive carcinoma compared to that in normal, benign, and borderline tumors. DKK3 loss occurred in 56.1% invasive carcinomas and was significantly associated with disease-free survival and chemoresistance in serous adenocarcinoma. DKK3 was lost in paclitaxel-resistant cells, while β-catenin and P-glycoprotein were upregulated. Exogenous secreted DKK3, incorporated by cells, enhanced anti-tumoral effect and paclitaxel susceptibility in paclitaxel-resistant cells, and reduced the levels of active β-catenin and its downstream P-glycoprotein, suggesting that DKK3 can be used as a therapeutic for targeting paclitaxel-resistant cancer.

## 1. Introduction

Ovarian cancer remains the most lethal gynecological malignancy worldwide. Current therapeutic approaches for treating epithelial ovarian cancer are relatively effective for the early stage; however, more than 60% of patients with ovarian cancer are diagnosed at an advanced stage (stage III or IV) with metastasis to distant organs beyond the pelvis [1,2]. Most patients with advanced epithelial ovarian cancer ultimately develop recurrent disease, despite achieving clinical remission after completing the initial treatment. In particular, the serous subtype accounts for nearly 70% of cases of epithelial ovarian cancer and has a poorer prognosis than the other major subtypes, including mucinous, endometroid, and clear cell carcinoma. Chemoresistance is considered the main cause of treatment failure [1,3]. Thus, to facilitate treatment of this subtype identification of therapeutic candidates that can overcome chemoresistance is critical.

Dickkopf-3 (DKK3) is considered a tumor suppressor as it is often deleted in cancers [4,5] and is frequently downregulated owing to epigenetic inactivation in cervical, lung, prostate, bladder, gallbladder, and breast cancers [6,7,8,9,10,11,12,13]. In addition, a significant relationship between aberrantly reduced DKK3 expression and poor prognosis was reported in uterine, cervical, colorectal, and pancreatic cancers [10,14,15,16,17]. Previous in vitro studies have shown that the tumor suppressor potential of DKK3 is mediated via angiogenesis in ovarian cancer cell lines and via mitochondrial and Fas death receptor pathways in mucinous ovarian cancer cells [18,19]. DKK3 mRNA expression in 35 of 56 tumors was lower than that in normal samples [20]. However, the clinical significance of DKK3 protein expression in serous ovarian cancer and its therapeutic role has not been elucidated.

P-glycoprotein, also known as multidrug resistance protein (MDR1) or ATP-binding cassette sub-family B member 1 (ABCB1), acts as an ATP-dependent efflux pump of the ABC transporter family, which transports a broad range of substrates, including paclitaxel, to outside the cell [21]. It is involved in the chemoresistance of various cancers [21,22,23,24] and has been a research target for several decades [25,26]. Its expression is promoted by the activation of β-catenin signaling [27]. However, the correlation of DKK3 and β-catenin-P-glycoprotein in ovarian cancer has not been reported yet.

Hence, in this study, we assessed DKK3 expression and its clinical significance using tissue blocks and the clinicopathological information of patients with serous ovarian cancer; finally, we evaluated the therapeutic function of DKK3. Here we report that DKK3 may be used as a biotherapeutic molecule to target paclitaxel-resistant ovarian cancer.

## 2. Materials and Methods

### 2.1. Patients

Tissue blocks from 82 patients with histologically proven invasive primary epithelial ovarian carcinoma were included in this study. Fifteen normal ovaries, 19 benign adenoma, and 10 borderline epithelial tumors were used as controls. The medical records and pathological findings of 42 patients with serous subtype were assessed for the clinicopathological parameters of age, serum CA125 level, Federation of Gynecology and Obstetrics (FIGO) stage, grade, debulking status, chemoresistance, and recurrence. Subjects with neoadjuvant chemotherapy were excluded. All patients underwent surgery as their primary treatment between 1995 and 2004 at the Department of Gynecologic Oncology at Chung-Ang University Hospital, Seoul, Korea. All patients underwent debulking surgery, including total abdominal hysterectomy, bilateral salpingo-oophorectomy, omentectomy, and lymph node dissection, and received minimum six cycles of postoperative chemotherapy with paclitaxel and carboplatin. Disease stage was assigned according to the International FIGO staging system, and the tumor grade and histological type were determined following World Health Organization standards. Grade 1 carcinomas were categorized to type I, and grade 2 and 3 carcinomas were categorized to type II [28]. Optimal cytoreduction was defined as less than 1 cm residual tumor after primary surgery. The study was approved by the institutional review board (approval number, C2015106 (1564)).

### 2.2. Immunohistochemical Analysis

Formalin-fixed and paraffin-embedded sections were cut to a thickness of 5 μm, dewaxed in xylene, and then dehydrated in alcohol gradient. After the slides were rinsed, endogenous peroxidase activity was blocked using 3% H_2_O_2_ for 15 min. The slides were pretreated with citrate buffer (10% citrate buffer stock in distilled water, pH 6.0) and autoclaved for 15 min to retrieve the antigen. Nonreactive blocking was performed with 1.0% horse serum in Tris-buffered saline (TBS), pH 6.0, for 3 min. Primary goat polyclonal antibody against human DKK3 (R&D Systems, Minneapolis, MN, USA), diluted 1:100 in phosphate-buffered saline (PBS) (pH 7.4), was applied and incubated for 1 h at 37 °C in a humidified chamber. A negative control was prepared by substituting the non-immune serum for primary antibody. After the slides were rinsed with PBS, they were incubated with secondary antibody for 10 min at room temperature (15–25 ℃) and then again rinsed in PBS. Antibody binding was detected using a standard labeled streptavidin-biotin system (Life Science Division, Mukilteo, WA, USA). The slides were dehydrated and mounted after counterstaining with Mayer’s hematoxylin. The immunoreactivity of DKK3 was scored for the tumor cells. DKK3 staining was evaluated for each tissue sample using a semi-quantitative staining intensity score (0, undetectable; 1+, weakly positive; 2+, moderate positive; 3+, intensely positive). One pathologist who was blind to the patients’ clinical histories reviewed all the cases.

### 2.3. Cell Culture

The 293 normal human embryonic kidney cells and TOV-21G and OV-90 ovarian cancer cell lines were purchased from the American Type Culture Collection (ATCC, Manassas, VA, USA). The cells were maintained in Eagle’s minimum essential medium (293 cells) and Roswell Park Memorial Institute (RPMI)-1640 medium (TOV-21G and OV-90) supplemented with 10% fetal bovine serum (FBS), 100 units/mL penicillin, and 100 μg/mL streptomycin (Invitrogen Corp, Carlsbad, CA, USA). All culture media and FBS were from Welgene (Daegu, Korea). The cells were incubated at 37 °C in a humidified atmosphere with 5% CO_2_. Media were routinely changed after every 3 d.

### 2.4. Establishment of Paclitaxel-Resistant Cell Lines

The resistant cell lines were established using a paclitaxel concentration gradient method. When the parent cell confluence reached 80%, the cells were treated with paclitaxel (Boryung Pharmaceutical, Seoul, Korea). Initially, the exponentially growing cells were exposed to the IC_50_ concentration of paclitaxel (10 ng/mL and 25 ng/mL for TOV-21G and OV-90, respectively) which are based on the results of the 3-(4,5-dimethylthiazol-2-yl)-2,5-diphenyltetrazolium bromide (MTT) assay. Each time paclitaxel was added to the cells, the dead cells were discarded after 48 h of treatment and the remaining viable cells were considered drug-resistant upon reaching 70–80% confluence. The remaining viable cells were then cultured in a higher concentration of paclitaxel and the above process was repeated. With gradual increase in paclitaxel concentration, the paclitaxel-resistant cell lines (TOV-21G/PTX and OV-90/PTX) survived in a final culture medium containing 0.1 μg/mL and 0.2 μg/mL paclitaxel, respectively.

### 2.5. Preparation of Recombinant Human DKK3

The human FLAG-DKK3 stable expression clone in TOV-21G cells was established using lipofectamine transfection of the p3X FLAG DKK3 plasmid encoding the full length human DKK3 gene with FLAG tag. TOV-21G cells stably expressing FLAG-DKK3 were cultured in FBS-free RPMI-1460 medium for 3 d before collecting the supernatant. The culture supernatant was concentrated using Amicon Ultra-15 centrifugal unit, NMWL 10,000 (Merck Millipore, Co. Ltd., Cork, Ireland), followed by FLAG immunoprecipitation to purify FLAG-DKK3 (Sigma-Aldrich, St. Louis, MO, USA). The purified human FLAG-DKK3 protein stock solution was maintained at −80 °C until use. Protein concentration was measured using the bicinchoninic acid protein assay (Thermo Scientific, Rockford, IL, USA).

### 2.6. Preparation of Conditioned Medium

DKK3 conditioned media was harvested from DKK3-transfected 293 cells. The cells were plated in a 6-well culture dish, transfected with 2 μg DKK3 plasmid (293-DKK3) or empty plasmid (293-Control), and incubated for 2 d (80–90% confluence). The culture media was harvested and Western blotting was performed to detect DKK3 protein in the culture media.

### 2.7. Three-Dimensional (3D) Spheroid Assay

OV-90 and OV-90/PTX cells were cultured in a microwell array as described before [29]. Briefly, the diameter of each well in the microwell was 600 μm, the depth was 600 μm, and the distance between the wells was 1.5 mm. To minimize cell loss, the entrance to the well was sloped. Polydimethylsiloxane (PDMS, Sylgard 184, Dow Corning, Midland, MI, USA), which has been widely used for cell culture because of its transparent and biocompatible characteristics, was used as the microplate material. The microwell array was coated with Pluronic F-127 (P2443, Sigma, St. Louis, MO, USA) to prevent cell attachment. Air bubbles trapped in the microwells were removed via manual pipetting. The next day, 5 × 10^5^ cell/mL were loaded in the microarray and approximately 3 × 10^3^ cells were present in each microwell. The cells aggregated after 20 min of seeding, indicative of spheroid formation; then, the ovarian cancer-forming spheroids were incubated with the conditioned media with or without PTX (200 ng/mL) and half of the medium was replaced every day. After 6 d, the spheroids in each well were imaged using light microscopy (magnification, 40×; scale bar 300 μm). Martin’s diameter, which indicates the distances between opposite edges passing through the center of the spheroid, was measured using the ImageJ software (NIH Image Processing and Analysis in Java), as the ovarian cancer-forming spheroids were not perfectly circular [30]. Spheroid diameter was finally decided as the mean value of several diameters in different orientations. Cell viability in spheroids was measured using the MTT assay after preparing single-cell suspension.

### 2.8. Immunofluorescence Analysis

The OV-90/PTX and TOV-21G/PTX cells were cultured in a cell culture dish at the density 1 × 10^4^ cells/cm^2^ and then incubated with DKK3 conditioned medium for 24 h. Subsequently, the cells were fixed with 4% formaldehyde for 15 min and incubated with a permeabilization buffer at 25 °C for 15 min. After blocking with 5% bovine serum albumin, the cells were co-stained with primary FLAG antibody (#ant-146, Prospec, East Brunswick, NJ, USA) and anti-non-phospho-β-catenin (#D13A1, Cell Signaling, Beverly, MA, USA) for 1 h, followed by washing twice with PBS. After incubation with a goat anti-mouse IgG-FITC secondary antibody and a goat anti-rabbit IgG-TR secondary antibody (sc-2010 and sc-2780, respectively, Santa Cruz Biotechnology, Santa Cruz, CA, USA) for 1 h at room temperature, the nucleus was stained using Hoechst 33,442 for 10 min. Fluorescence signals were observed using fluorescence microscopy.

### 2.9. Western Blot Analysis

For Western blot analysis, the cells were lysed for 30 min on ice in radioimmunoprecipitation assay buffer with a protease inhibitor cocktail (P8340, Sigma, USA), and the lysates were cleared via centrifugation at 14,000 rpm for 15 min. The proteins were separated on 8–10% sodium dodecyl sulfate polyacrylamide gel and electro-transferred to a Hybond ECL nitrocellulose membrane (Amersham Pharmacia Biotech). The membrane was then blocked with 3% bovine serum albumin in 1× TBS containing 0.1% Tween 20 for 4 h and incubated overnight at 4 °C with appropriate dilution of each primary antibody, according to antibody datasheet. The membranes were rinsed thrice for 15 min each with washing buffer (1× TBS containing 0.1% Tween 20) and incubated with the appropriate secondary antibody at room temperature for 1 h. After rinsing thrice for 15 min with washing buffer at room temperature, the membrane was developed using SuperSignal West Pico PLUS chemiluminescent substrate (Thermo Scientific, Meridian Rd., Rockford, IL, USA). The protein bands in the Western blot films were quantified using the NIH ImageJ software. The experiment was performed in triplicate and repeated at least three times.

### 2.10. MTT Assay

MTT (3-(4,5-Dimethylthiazol-2-yl)-2,5-Diphenyltetrazolium Bromide) reagent (12 mM) was added to the cells and incubated for 3 h according to the manufacturer’s protocol (M6494, Invitrogen, Eugene, OR, USA). At the end of the incubation, 100 μL dimethyl sulfoxide was added to dissolve the MTT reaction product, formazan, at room temperature for 30 min. Then, cell density was estimated as absorbance at 540 nm using an enzyme-linked immunosorbent assay (ELISA) reader (Epoch2, BioTek, Winooski, VT, USA). The experiment was performed in triplicate and repeated at least three times.

### 2.11. Migration Assay

Cells were seeded in a 6-well plate (SPL Life Science, Gyeonggi-do 487-835, Korea) at 80–90% confluence. After 24 h, the cells were treated in different ways and the monolayer was scratched using a 1000 μL tip. The cells were incubated at 37 °C in a humidified atmosphere with 5% CO_2_. Cell motility was tracked 24 h post-treatment. The migration rate was calculated as the percentage of recovered area compared to the initial gap at 0 h. Images were captured using a light microscope (magnification 10×; scale bar, 200 μm). The mean gap area was calculated using the ImageJ software. The experiment was performed in triplicate.

### 2.12. Chemicals and Antibodies

Paclitaxel was obtained from Boryung Pharmaceutical (Seoul, Korea). Lithium chloride (LiCl) was purchased from Sigma (L9650, MO, USA). The following antibodies were used for the Western blot analysis: anti-FLAG from Prospec (USA), anti-β-actin from Santa Cruz Biotechnology (Santa Cruz Biotechnology, Santa Cruz, CA, USA), anti-DKK3 from R&D Systems (Minneapolis, MN, USA), anti-β-catenin and anti-non-phospho-β-catenin from Cell Signaling (Beverly, MA, USA), anti-P-glycoprotein from Invitrogen, Thermo Fisher (Waltham, MA, USA), and anti-E-cadherin from Santa Cruz Biotechnology.

### 2.13. Statistical Analysis

Survival was estimated using Kaplan–Meier estimates and compared using the log-rank test. Median counts were analyzed using the Mann–Whitney U-test. Kruskal–Wallis test was used to compare DKK3 expression among normal, benign, borderline, and invasive tumors. Dichotomous groupings were analyzed using the Chi-square and Fisher’s exact test as appropriate. Multivariate analysis was performed using the Cox regression method. The Mann–Whitney U-test was used to compare values between two groups. Cell viability was compared between two groups using the Mann–Whitney U-test with Bonferroni correction. All *p*-values reported are two-sided, and statistical significance was defined as *p* < 0.05. Statistical analysis was performed using the Statistical Package for the Social Sciences (SPSS, version 15.0, Chicago, IL, USA).

## 3. Results

### 3.1. Aberrant Downregulation of DKK3 in Invasive Ovarian Carcinoma

Immunohistochemical analysis showed that DKK3, present predominantly in the cytoplasm of cells (Figure 1a,b), was abundantly expressed in normal ovarian epithelial cells and benign adenoma. DKK3 expression was significantly lower in invasive cancer tissues than in normal, benign, and borderline tumors (Table 1), while it did not differ among normal, benign, and borderline tumors. Approximately 56.1% (46 of 82) of invasive carcinoma tissues showed complete loss of DKK3; DKK3 was absent in most mucinous, serous, endometrioid, and undifferentiated carcinomas, while its expression was not reduced in transitional cell carcinoma.

### 3.2. Characteristics and Survival Analysis of 42 Patients with Serous Ovarian Carcinoma

Analysis of data from all 82 patients with invasive carcinoma revealed that DKK3 expression was not significantly associated with survival and clinicopathological parameters (Appendix A). As histological type is one of the main factors used in predicting prognosis in patients with ovarian cancer, 42 patients with serous adenocarcinomas were selected for further evaluation. Twenty-six of forty-two (61.9%) patients exhibited DKK3 loss (Table 1). The clinicopathological parameters of these patients are listed in Table 2. After a median follow-up of 27.5 months (range, 1–151 months), 25 (59.5%) patients did not show evidence of disease, and 17 (40.5%) patients experienced recurrent disease. Disease-free survival was defined as the length of time (months) from the last therapy to the diagnosis of the first recurrence. Univariate survival analysis showed that the FIGO stage, DKK3 protein expression, surgical debulking optimality, and chemo-response were significant factors affecting disease-free survival (Figure 1c). Multivariate survival analysis adjusted for FIGO stage, surgical debulking optimality, chemo-response, and DKK3 expression showed that women with chemosensitivity were at significantly lower risk of recurrence than women with chemoresistance (Table 2, *p* < 0.01).

### 3.3. DKK3 Loss Was Significantly Associated with Chemoresistant Ovarian Cancer

The 42 patients with serous adenocarcinoma were stratified by the absence or presence of the DKK3 protein into negative and positive groups. The clinicopathological characteristics of the patients in each group are summarized in Table 3 and statistical analysis showed a significant association between DKK3 protein loss and chemoresistance (*p* = 0.029). There was no difference in DKK3 expression according to age, FIGO stage, surgical debulking optimality, preoperative serum level of CA-125, and type or disease recurrence between the two groups.

### 3.4. DKK3 Was Lost in Paclitaxel-Resistant Ovarian Cancer Cells

Paclitaxel-resistant cells from TOV-21G and OV-90 were established using a paclitaxel concentration gradient method (Figure 2a,b). These paclitaxel-resistant cell lines did not show any morphological changes and their growth rates were similar to those of parental cell lines. The paclitaxel-resistant cells proliferated persistently when the cells were treated with paclitaxel, while the viability of the parental cells decreased markedly (Figure 2c,d).

As DKK3 loss was significantly associated with chemoresistance of patients with ovarian cancer, we assessed DKK3 protein level in the paclitaxel-resistant cells. Western blot analysis showed that, compared to that in parental cells, DKK3 expression was markedly reduced and beta-catenin was upregulated in paclitaxel-resistant cells. P-glycoprotein was upregulated in both paclitaxel-resistant cell lines, suggesting a link between P-glycoprotein level and DKK3 loss (Figure 2e and Appendix A).

### 3.5. Secreted DKK3 Enhanced Paclitaxel Susceptibility of Ovarian Cancer Cells

DKK3 is a well-known secreted protein [31]. Therefore, we treated the cells with conditioned media containing secreted DKK3, which was produced by DKK3-overexpressing 293 cells. The media of empty vector-transfected cells was used as the control media. The proliferation of TOV-21G and TOV-21G/PTX was inhibited by treatment with DKK3-conditioned media in a dose-dependent manner, while the cells grew well in the control media (Figure 3a,b). The viability of OV-90 cells increased in the control media but decreased in the -conditioned media, while OV-90/PTX cells were inhibited by both the conditioned media, with the inhibitory effect of the DKK3-conditioned media being stronger than that of the control media (Figure 3c). Next, we tested the effect of purified recombinant human DKK3 protein in OV-90 and OV-90/PTX cells and found a dose-dependent inhibition in both cell lines (Figure 3d). We further assessed whether secreted DKK3 increased the anti-tumoral effect of paclitaxel in paclitaxel-resistant cells. Cotreatment with paclitaxel and control conditioned media did not interrupt the growth of TOV-21G/PTX and OV-90/PTX cells; however, cotreatment with paclitaxel and DKK3-conditioned media significantly decreased the viability of both paclitaxel-resistant cell lines (Figure 3e,f), suggesting synergism between the effects of DKK3 and paclitaxel. Next, the anti-proliferative effect of DKK3 was evaluated in three-dimensional (3D) tumor spheroids of OV-90 and OV-90/PTX formed in the microwell arrays (Figure 3g,h). We observed that DKK3 CM treatment shrunk the spheroids and that cotreatment with DKK3 CM and paclitaxel disassembled them, indicative of serious damage. We compared the diameter of the spheroids to evaluate the effect of the treatments on spheroid growth, as the clinical response of cancer treatment was previously evaluated by measuring tumor diameter [32]. Simultaneously, we assessed cellular viability using the MTT assay after suspension of spheroids into single cells. The size and viability of spheroids in the DKK3 conditioned medium were significantly lower than those in the control conditioned medium. Moreover, cotreatment of OV-90/PTX with secreted DKK3 and paclitaxel synergistically reduced spheroid size despite no difference in cell viability. Next, the anti-tumoral effect of DKK3 on OV-90/PTX and TOV-21G/PTX was evaluated using migration assay (Figure 3i,j). Cell migration in DKK3 CM was remarkably inhibited compared to that in the control CM, and cotreatment with DKK3 CM and paclitaxel synergistically reduced cell motility. Furthermore, to understand the underlying mechanism, we assessed the expression of E-cadherin, an epithelial marker. Loss of E-cadherin expression reflects cancer metastasis [33]. Our results showed that E-cadherin was upregulated in the DKK3 CM group compared to that in the control CM group, indicating that DKK3 inhibits ovarian cancer cell migration by restoring epithelial–mesenchymal transition (Figure 3k,l and Appendix A). Therefore, secreted DKK3 exerted an anti-tumoral effect in paclitaxel-resistant ovarian cancer cells and enhanced paclitaxel susceptibility.

### 3.6. Secreted DKK3 Enhanced Paclitaxel Susceptibility of Ovarian Cancer Cells via Inhibition of β-Catenin-P-Glycoprotein Signaling Pathway

As DKK3 loss is associated with chemoresistance and upregulation of the active form of β-catenin and P-glycoprotein, we investigated the relationship between them. We first assessed whether the secreted DKK3 was internalized by the cells. Immunofluorescence analysis showed perinuclear accumulation of FLAG-DKK3 and an inverse correlation between DKK3 and the active form of β-catenin in OV-90/PTX and TOV-21G/PTX cells (Figure 4a,b). After 24 h of incubation, secreted DKK3 reduced the levels of the active form of β-catenin and P-glycoprotein (Figure 4c,d, Appendix A). We then activated β-catenin with LiCl, a glycogen synthase kinase (GSK) inhibitor, in paclitaxel-resistant cells. Western blotting showed that activated β-catenin upregulated P-glycoprotein in chemoresistant cells treated with a control conditioned medium. However, β-catenin was not activated in the chemoresistant cells treated with the DKK3 conditioned medium, and accordingly, P-glycoprotein was not upregulated (Figure 4e,f, Appendix A). On the basis of these results, we concluded that secreted DKK3 reduced the level of P-glycoprotein, one of the most important chemoresistance-regulating proteins, by inhibiting β-catenin.

## 4. Discussion

This study has assessed the clinical significance of DKK3 loss and its role as a potential biotherapeutic molecule in ovarian cancer. In this study, we found that DKK3 was significantly downregulated in invasive epithelial ovarian carcinoma compared to that in normal tissue, adenoma, and borderline tumors. Moreover, complete loss of DKK3 was observed in 56.1% of cases of invasive epithelial ovarian carcinomas. Taken together with the results of a previous report showing the absence of DKK3 in 63% cases of 56 ovarian carcinoma tissue samples [20], we concluded that >50% invasive epithelial ovarian cancers lose DKK3 during carcinogenesis. DKK3 is frequently downregulated in various cancers [6,10,14,16,17,18,34,35,36], and therefore, this study supports the hypothesis that downregulation of DKK3 occurs commonly during carcinogenesis.

In this study, we analyzed the relationship between DKK3 expression and the clinicopathological factors and prognosis of patients with serous ovarian cancer. Although a previous study did not find any correlation between DKK3 loss and clinicopathological parameters of ovarian cancer [20], the results of univariate survival analysis in this study showed that DKK3 loss was a prognostic factor, along with the FIGO stage, suboptimality of debulking surgery, and chemoresistance. This may be explained by the variation in the clinical behaviors of different histological ovarian cancers [1,2]. We analyzed the clinicopathological parameters of only the serous type in this study, although a previous study has included all types. Multivariate analysis showed that only chemo-response was an independent prognostic factor for patients with serous ovarian carcinoma. Because of the strong correlation between DKK3 loss and chemoresistance, the significance of DKK3 loss was nullified after adjustment. Despite the limitations associated with the retrospective nature and small sample size of this study, this is the first report to show the clinical significance of DKK3 loss in ovarian cancer. A significant correlation between DKK3 downregulation and poor prognosis was reported in patients with cervical squamous cell carcinoma or gastric cancer [10,16], and DKK3 loss was significantly associated with higher Gleason scores in prostate cancer [37]. In addition, compared to that observed in patients with strong DKK3 expression, DKK3 downregulation in tumors of gastric cancer tissue samples was associated with short disease-free survival [38]. Taken together, these findings suggest that aberrant reduction in DKK3 expression is a clinical biomarker for identifying high-risk cancer patients with poor prognoses.

As a secreted glycoprotein, DKK3 can be detected in patient sera using ELISA. In fact, the serum level of DKK3 in patients with ovarian cancer was found to be lower than that in normal individuals [39]. These findings prompted us to assess the anti-tumor activity of DKK3 secreted in the medium in a monolayer culture system and 3D spheroid model. We observed endocytosis of secreted DKK3 6 h after treatment, which showed anti-proliferative activity, as endogenous DKK3 is downregulated in cancer cells and lost in paclitaxel-resistant cells. Moreover, the cytotoxicity of paclitaxel on paclitaxel-resistant cells was restored when treated with secreted DKK3, but not with the control medium, indicating that secreted DKK3 restored paclitaxel susceptibility. As the 3D spheroid structure mimics the physical and biochemical features of a natural solid tumor mass, it may be similar to cells growing in vivo. Thus, our result suggested that secreted DKK3 may effectively penetrate the tumor mass, resulting in fractional killing, which is consistent with the results of previous reports showing that an adenovirus carrying DKK3 induced apoptosis in bladder, prostate, biliary, and liver cancers [34,35,40,41,42]. Taken together, we believe that DKK3 may be a promising cancer therapeutic for overcoming drug resistance, which may be utilized in various types of biopharmaceutics.

In our paclitaxel-resistant ovarian cancer cells, DKK3 was downregulated and P-glycoprotein was upregulated. Interestingly, when DKK3 was re-expressed after treatment with secreted DKK3, P-glycoprotein expression was attenuated. So far, the signaling link between DKK3 and P-glycoprotein has not been elucidated. P-glycoprotein, acting as an ATP-dependent efflux pump, transports various chemo agents, such as paclitaxel, to outside the cells, which is a well-known mechanism of drug resistance [21,22,23,43]. Previous studies have shown that adenovirus-DKK3 treatment suppressed P-glycoprotein expression in an Akt/NFκB or c-Jun-kinase (JNK)-dependent manner [8,44,45]. In addition, we have previously shown that DKK3 inactivated β-catenin by direct interaction with β-TrCP, which was reported to be involved in β-catenin degradation [15,46,47]. Here, we established a signaling link among DKK3, β-catenin, and P-glycoprotein, in which DKK3 negatively regulates P-glycoprotein by inhibiting β-catenin. In addition, activation of β-catenin by LiCl, a β-catenin signaling agonist, resulted in P-glycoprotein upregulation. These results indicated that β-catenin is an upstream regulator of P-glycoprotein. Moreover, P-glycoprotein was downregulated after treatment with secreted DKK3. Taken together, the chemosensitizing effect of secreted DKK3 was mediated via a downregulation of β-catenin-P-glycoprotein signaling. Therefore, negative regulation of P-glycoprotein expression may be a mechanism via which DKK3 restores paclitaxel susceptibility.

As chemoresistance is an obstacle in ovarian cancer therapy [1,2], the anti-tumoral effect of DKK3, as well as its synergistic effect with paclitaxel in paclitaxel-resistant ovarian cells, indicates the possibility of a new therapeutic strategy. The subgroups of four patients with chemoresistance/expressing DKK3 and nine patients with chemosensitivity/lacking DKK3 could not be explained. We investigated one of the various mechanisms functioning in individual cancer cells; therefore, our findings are not applicable to all ovarian cancers. Patients who may benefit from DKK3 therapy should be individually identified. Moreover, as the mechanism of action of DKK3 involves regulation of P-glycoprotein expression, its anti-tumoral effect may not be limited to only paclitaxel-resistant cells. However, further work in this direction is warranted.

## 5. Conclusions

In conclusion, DKK3 was significantly downregulated in invasive epithelial ovarian carcinomas compared to that in normal, benign, and borderline tumors. DKK3 loss was more frequent in serous ovarian cancers with chemoresistance than in those with chemosensitivity. Moreover, DKK3 exerted an anti-proliferative effect on ovarian cancer cells and reversed paclitaxel resistance by downregulating P-glycoprotein via inhibition of β-catenin. Therefore, secreted DKK3 may be a potential drug to target drug-resistant cancers such as epithelial ovarian cancer. Hence, developing human secreted DKK3 as a biotherapeutic molecule, in cancer treatment, will be added in our future work.

## Figures and Tables

**Figure 1 cancers-14-00924-f001:**
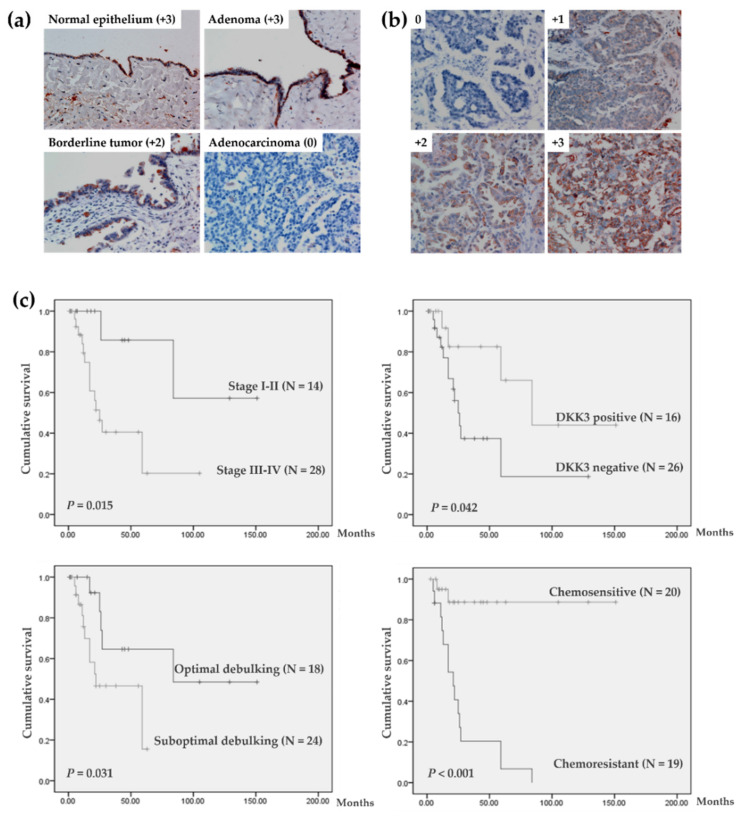
Immunoreactivity of DKK3 and survival curves for patients with invasive serous ovarian cancer. (**a**) DKK3 immunoreactivity in normal epithelium and serous tumors. (**b**) Immunoreactivity of each score in invasive serous carcinoma. (**c**) Kaplan–Meier curves for disease-free survival. Magnification 200×.

**Figure 2 cancers-14-00924-f002:**
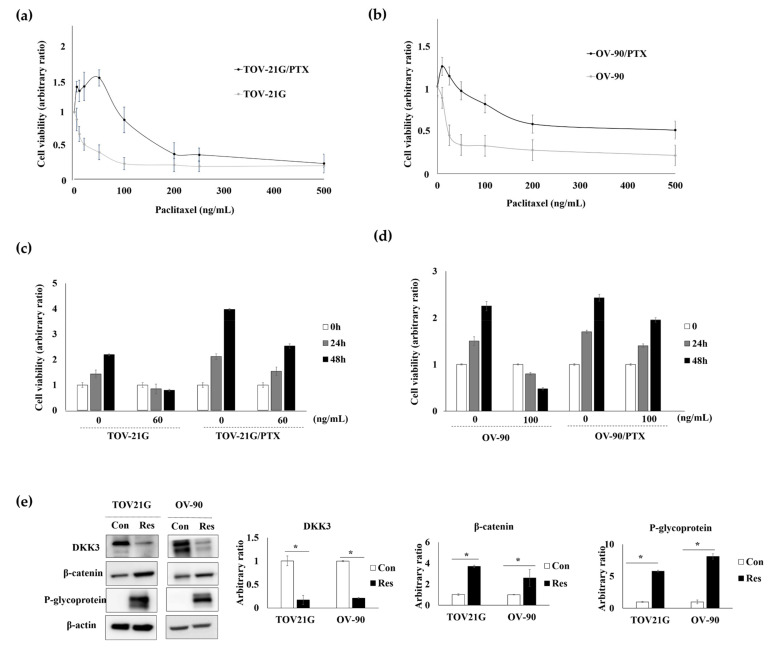
Characteristics of paclitaxel-resistant cells. (**a**,**b**) Cells were seeded in triplicates in 96-well plates in 0.1 mL culture medium at a density of 1 × 10^4^ cells/well. After 24 h, the cells were treated with paclitaxel. MTT assay performed 48 h after treatment showed that the paclitaxel-resistant cells (TOV-21G/PTX and OV-90/PTX) had higher IC_50_ than the parental cells. (**c**,**d**) Paclitaxel treatment reduced the viability of parental cells, while the paclitaxel-resistant cells continued to grow. (**e**) Western blot analysis revealed that DKK3 was lost and that β-catenin and P-glycoprotein were upregulated in paclitaxel-resistant cells (Res) compared to that in their parent cells (Con). β-Actin was used as the loading control. * *p* < 0.01.

**Figure 3 cancers-14-00924-f003:**
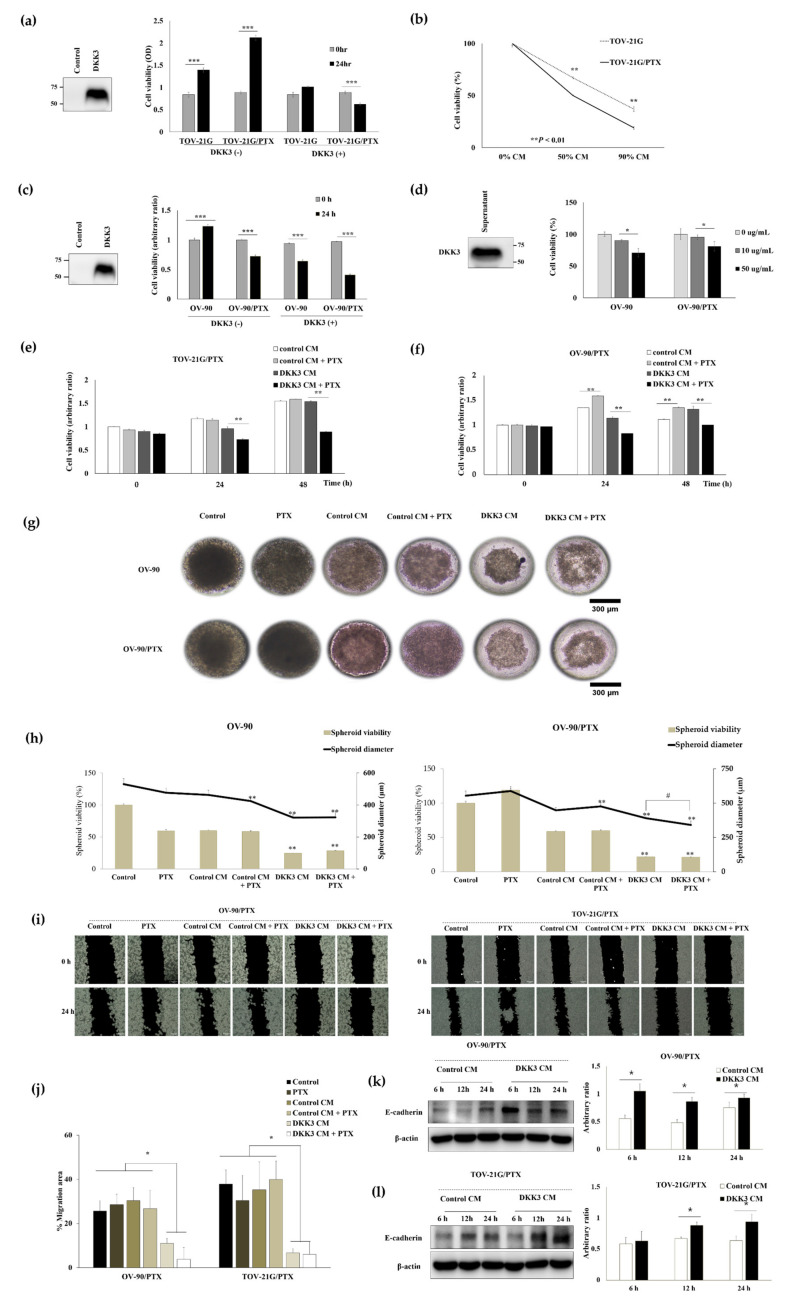
Anti-proliferative effect of the secreted DKK3 on paclitaxel-resistant cells. (**a**) Western blotting results show DKK3 protein levels in the control and DKK3 conditioned medium (CM). MTT assay performed after 24 h of incubation with CM (*** *p* < 0.001). (**b**) The viability of TOV-21G and TOV-21G/PTX cells after treatment with DKK3 CM diluted to 0%, 50%, and 90%, respectively, with control CM (** *p* < 0.01). (**c**) The OV-90 and OV-90/PTX cells were incubated with CM with or without DKK3. Western blotting results show DKK3 protein levels in the control and DKK3 CM. MTT assay after 24 h of treatment (*** *p* < 0.001). (**d**) Western blotting reveals recombinant human DKK3 levels. Two concentrations of the protein (10 μg/mL and 50 μg/mL) were used to treat cells for 72 h (* *p* < 0.05). (**e**,**f**) TOV-21G/PTX and OV-90/PTX cells were treated with CM with or without paclitaxel (100 ng/mL and 200 ng/mL, respectively) (** *p* < 0.01). (**g**) 3D spheroids of OV-90 and OV-90/PTX cells were generated in a microwell array and the images were captured after incubation for 6 d (magnification, 40×; scale bar 300 μm). (**h**) Spheroid viabilities and diameters were compared. The asterisks on the graph denote statistically significant differences (*p* < 0.01) between the DKK3 CM group and the control CM group. The diameter of spheroids in the DKK3 CM group was compared to that in the DKK3 CM with paclitaxel group (#, *p* < 0.01). (**i**,**j**) OV-90/PTX and TOV-21G/PTX cells were treated as mentioned above and migration rates were evaluated using the migration assay. (**k**,**l**) The paclitaxel-resistant cells were incubated with control and DKK3 CMs. At the indicated time points, the cells were harvested and subjected to Western blotting. The graphs were plotted based on the band densities measured using the ImageJ software. * *p* < 0.05.

**Figure 4 cancers-14-00924-f004:**
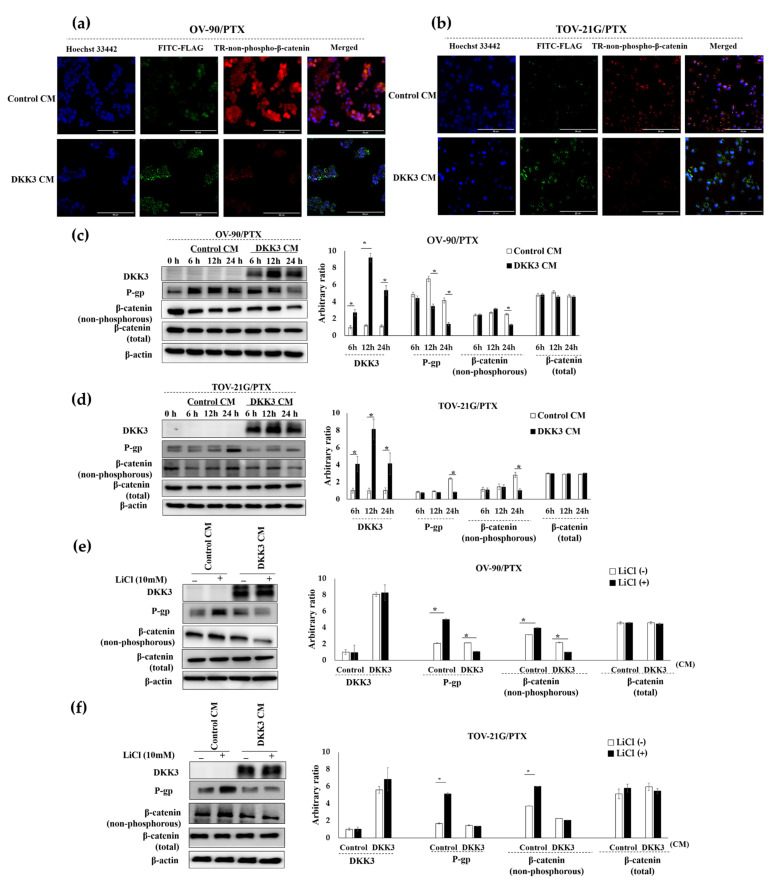
Secreted DKK3 enhanced paclitaxel susceptibility via inhibition of the β-catenin-P-glycoprotein signaling pathway. (**a**,**b**) The cells were incubated with control and DKK3 CM. Immunofluorescence analysis after 24 h showed that FLAG-DKK3 was present in the perinuclear area, attenuating TR-non-phospho-β-catenin signaling. Scale bar, 50 μm. (**c**,**d**) The cells were incubated with control and DKK3 CM and subjected to Western blotting. The graphs were plotted based on the band densities (* *p* < 0.01). (**e**,**f**) To activate endogenous β-catenin, cells were treated with LiCl for 24 h and subjected to Western blot analysis. β-Actin was used as the loading control. The intensities of the Western blot bands were normalized to that of β-actin for comparison (* *p* < 0.01).

**Table 1 cancers-14-00924-t001:** Expression of DKK3 in different ovarian tissue samples.

	Immunoreactivity, N (%)					
	0	+1	+2	+3	Total	*p*-Value
Normal epithelium	0	2 (13.3)	6 (40)	7 (46.7)	15 (100)	<0.001 *
Benign adenoma	0	4 (21.1)	8 (42.1)	7 (36.8)	19 (100)	
Borderline tumor	2 (20.0)	5 (50.0)	2 (20.0)	1 (10.0)	10 (100)	
Invasive carcinoma	46 (56.1)	16 (19.5)	6 (7.3)	14 (17.1)	82 (100)	
Mucinous	10	0	1	2	13	
Serous	26	11	4	1	42	
Endometrioid	3	2	0	0	5	
Transitional cell	0	1	0	11	12	
Clear cell	1	1	1	0	3	
Undifferentiated	6	1	0	0	7	

* Kruskal–Wallis test was used to compare DKK3 expression among normal, benign, borderline, and invasive tumors.

**Table 2 cancers-14-00924-t002:** Clinicopathological parameters and disease-free survival analysis of prognostic factors in 42 patients with serous adenocarcinoma.

Clinicopathological Parameters	N (%)	*p*-Value	
		Univariate	Multivariate
Age			
Mean (range), year	53.2 (24–77)		
CA125			
≤35 U/mL	3 (7.1)		
>35 U/mL	39 (92.9)		
FIGO stage		0.015	* NS
I–II	14 (33.3)		
III–IV	28 (66.7)		
Type		NS	
I	12 (28.6)		
II	30 (71.4)		
DKK3 protein expression		0.042	NS
Negative	26 (61.9)		
Positive	16 (38.1)		
Debulking surgery		0.031	NS
Optimal	18 (42.9)		
Suboptimal	24 (57.1)		
Chemo-response		<0.001	<0.01
Sensitive	20 (47.6)		
Resistant	19 (45.2)		
Unknown	3 (7.2)		

* NS, no significance.

**Table 3 cancers-14-00924-t003:** Clinicopathological characteristics of women with/without DKK3 expression.

Clinicopathological Parameters	DKK3 Expression		
	Negative (*n* = 26)	Positive (*n* = 16)	*p*-Value
Age			* NS
Mean (±SD), year	52.9 (±11.8)	54.3 (±12.5)	
CA125			NS
≤35 U/mL	1	2	
>35 U/mL	25	14	
FIGO stage			NS
I–II	7	7	
III–IV	19	9	
Type			NS
I	7	5	
II	19	11	
Debulking operation			NS
Optimal	10	8	
Suboptimal	16	8	
Chemo-response			0.029
Sensitive	9	11	
Resistant	15	4	
Unknown	2	1	
Recurrence			NS
No	13	12	
Yes	13	4	

* NS, no significance.

## Data Availability

The data presented in this study are available in this article (and Appendix A).

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
