# Peer review of "DKK3, Downregulated in Invasive Epithelial Ovarian Cancer, Is Associated with Chemoresistance and Enhanced Paclitaxel Susceptibility via Inhibition of the β-Catenin-P-Glycoprotein Signaling Pathway"

_cancers, 2022, doi:10.3390/cancers14040924_

Round 1

Reviewer 1 Report

Major comment:

  1. Author showed the internalization and peri-nuclear accumulation of FLAG-DKK3 into the cells by Immunofluorescence in OV-90/PTX and TOV-21G/PTX cells. In addition, It would be important to show the inverse correlation of DKK3 and β-catenin by co-staining would strengthen the results and give more scientific insights to this manuscript.
  1. Author assessed the anti-proliferative effects of secterory DKK3 in TOV-21G and TOV-21G/PTX cells. In addition to the anti-proliferative effects, the other important characteristics of cancer cells such as migration and invasion should be analyzed in the same condition would give convincing evidence to confirm the anti-tumoral effects of DKK3.
  2. Author claimed that DKK3 may be a potential therapeutic that can target chemoresistance by reducing the level of P-glycoprotein, one of most important chemoresistance-regulating proteins, by inhibiting β-catenin. However, the molecular mechanism by which DKK3 regulate β-catenin is not shown.

Minor comment:

  1. Is there any reported data available to show the secretory levels of DKK3 in cancer patients?

Author Response

Thanks a lot.

Reviewer 2 Report

Nguyen et al. present a comprehensive study implicating the significance of DKK3 in serous ovarian cancer by first analyzing clinical samples followed by in vitro molecular characterization using two human cell lines. The manuscript is well written and clear with no major grammar issues noted. The study design is logical and the findings are impactful to the field yet clarifications in some sections, issues in presentation of the results, as well as additional details denoting rigor in the in vitro data are required prior to publication.

Introduction

While the introduction provides adequately cited research pertinent to the study, further elaboration on the standard of care treatment and its associated effectiveness would aide in understanding the impact of this work. Serous ovarian cancer is delineated as a specific subtype of interest yet no context for why this is important (incidence, classification, origin) is presented. Additionally, relevance of B-catenin and P-glycoprotein should also be addressed. Line 64: Please elaborate on the significance or details of this finding. In the concluding paragraph, can additional details be included to denote the evaluation of DKK3 as a therapeutic was performed in vitro and the primary findings.

Materials and methods

Please denote replicates and pertinent reproducibility information to insure rigor in the data in the methods or in results.

Results

All figure legends are too descriptive with much of the text belonging in the text (in methods and/or results) instead of in legend.

Figure 1. It is unclear why only serous carcinomas are selected for survival analysis when all invasive forms presented display DKK3 reduction in Figure 1 and Table 2.

Table 3. Why is the stratification system changed from the immunoreactivity scoring presented in Table 1?
Figure 2. How clinically relevant are the cell lines used and what subtype of ovarian cancer are these derived from? IC50 values were referenced however values are not presented for all cell lines. The presentation of MTT data needs to be standardized (y axis labeling of Figure 1a-d)
Figure 3. 3a and 3c blots bands are not clear and in line with other western blot data and no supplementary figure depicting whole blot was presented in supplementary files. The non-uninform assays performed on independent cell lines is confusing. Is there a rationale behind the selection of secreted media vs recombinant assays for only one of the two cell lines in Figure 3b and 3d? The 24 hour no PTX data presented in Figure 3e and d do not match 24 hour data presented in Figure 3a and 3c. Can you address this discrepancy in DKK3 effects without PTX? The synergistic effects of cotreatment shows only minor decrease in spheroid diameter in OV-90X yet it should be noted that viability is not significantly altered.

Figure 4. While the data presented is convincing to correlate P-glycoprotein to B-catenin activation, the final conclusion that this is the causation of chemoresistance is unsubstantiated in this study as no functional analysis is presented on P-glycoprotein in this setting so this statement should be hypothesized in discussion or removed from results.

Discussion

Please address the presence of resistant DKK3+ and sensitive DKK3- cancers and the potential impact on your in vitro findings.

DKK3 is classically characterized as a canonical Wnt antagonist and has been implicated in ovarian cancer and chemoresistance, yet this is not examined or mentioned at all throughout this manuscript. How are you confident that P-glycoprotein instead of Wnt signaling activation is mechanism of action for DKK3’s role in ovarian cancer aggressiveness?

In order to treat late stage cancer, systemic administration will be required. Is this a viable candidate for treatment. What is known of the pleiotropic effects of DKK3 be systemic administration?

Conclusion: reference to platinum resistance is not mentioned elsewhere in text

Author Response

Thanks a lot.

Reviewer 3 Report

Overall this is an interesting study. There are a few minor issues that has to be addressed before acceptance.

  1. Figure 1-a-b; Please also provide the Immunoreactivity score for IHC quantification.
  2. Figure 1-c; Please mention patient numbers (n=?) in km plotter data.
  3. Figure-3-a and c; Please repeat the western blot of DKK3. It’s not acceptable due to smearing pattern of DKK3 band.
  4. Figure-3-g; Please repeat the 3D spheroid formation assay. It’s not clear and unable to compare the result between different experimental condition.
  5. Figure-3-g; please quantitate 3D spheroid data in terms of number and size of spheroids.
  6. Please also provide some EMT related data (such as migration and invasion assay) with western blot of EMT markers such as E-Cadherin , N-Cadherin, Vimentin etc.
  7. Schema requires  some improvement for complete representation of hypothesis.
  8. Please include scale bars in all figures. 

Author Response

Thanks a lot.

Round 2

Reviewer 1 Report

The authors addressed all the comments adequately

Reviewer 2 Report

Thank you for your thoroughness in addressing the previous comments. The revisions/responses are appropriate and clarified the rationale for the presentation of data. I have no further revisions required. Good job!